# BMP-2 Gene Delivery-Based Bone Regeneration in Dentistry

**DOI:** 10.3390/pharmaceutics11080393

**Published:** 2019-08-05

**Authors:** Shin-Young Park, Kyoung-Hwa Kim, Sungtae Kim, Yong-Moo Lee, Yang-Jo Seol

**Affiliations:** 1Program of Clinical Dental Education and Dental Research Institute, School of Dentistry, Seoul National University, Seoul 03080, Korea; 2Predoctoral Treatment Center, Seoul National University Dental Hospital, Seoul 03080, Korea; 3Department of Periodontology and Dental Research Institute, School of Dentistry, Seoul National University, Seoul 03080, Korea

**Keywords:** bone morphogenetic protein 2, gene transfer technique, bone regeneration, animal experimentation

## Abstract

Bone morphogenetic protein-2 (BMP-2) is a potent growth factor affecting bone formation. While recombinant human BMP-2 (rhBMP-2) has been commercially available in cases of non-union fracture and spinal fusion in orthopaedics, it has also been applied to improve bone regeneration in challenging cases requiring dental implant treatment. However, complications related to an initially high dosage for maintaining an effective physiological concentration at the defect site have been reported, although an effective and safe rhBMP-2 dosage for bone regeneration has not yet been determined. In contrast to protein delivery, BMP-2 gene transfer into the defect site induces BMP-2 synthesis in vivo and leads to secretion for weeks to months, depending on the vector, at a concentration of nanograms per milliliter. BMP-2 gene delivery is advantageous for bone wound healing process in terms of dosage and duration. However, safety concerns related to viral vectors are one of the hurdles that need to be overcome for gene delivery to be used in clinical practice. Recently, commercially available gene therapy has been introduced in orthopedics, and clinical trials in dentistry have been ongoing. This review examines the application of BMP-2 gene therapy for bone regeneration in the oral and maxillofacial regions and discusses future perspectives of BMP-2 gene therapy in dentistry.

## 1. Introduction

Recently, the adjunctive use of bone morphogenic protein 2 (BMP-2) has become clinically available to improve bone regeneration and produce predictable results in challenging cases. BMP-2, first isolated by Urist et al. in 1965, is one of the potential growth factors inducing osteogenic differentiation and bone formation [1]. In 2007, the US Food and Drug Administration approved the use of recombinant human BMP-2 (rhBMP-2) in dentistry for maxillary sinus grafting and bone-grafting procedures associated with extraction sockets [2]. rhBMP-2 is delivered into a defect with synthetic bone-grafting materials (e.g., β-tricalcium phosphate) or a collagen sponge for dental use.

Drug delivery systems are critical for drugs, especially bioactive molecules, to function. Because the active period of each bioactive molecule is unique and dynamic, the appropriate delivery system for each bioactive molecule has significant effects in the clinic. For example, initially acting molecules should be delivered by a rapidly acting system, while slow acting molecules for wound healing need to be delivered by a system that facilitates sustained release over a certain period.

Gene delivery is one option for achieving the sustained release of target molecules during the healing period. Gene delivery involves transferring a target gene encoding BMP-2 into the defect site using vectors carrying the gene. Then, the cells transfected by vectors carrying the gene produce the target molecules in vivo, secrete the target molecules into the defect site. The drug release period can be controlled by the vector carrying the gene. In addition, the target molecules synthesized in vivo are effective because post-translational modifications and folding processes occur in the host cells, unlike the production of rhBMP-2 using *E. coli*.

In this article, we will review the role of BMP-2 in the wound healing process of bone tissue and the application of rhBMP-2 in dentistry. Additionally, BMP-2 gene delivery for oral and maxillofacial regions will be reviewed, and future perspectives of BMP-2 gene delivery for dental fields will be discussed.

## 2. BMPs in Bone Tissue Healing

A representative model for observing the bone tissue healing process in dentistry is the extraction socket-healing model. After tooth extraction, a series of alveolar bone wound healing processes take place for several months (Figure 1a) and can be divided into three phases, according to Araujo et al.—i.e., the inflammatory, proliferative, and modelling/remodelling phases [3,4]. First, the extraction socket is filled with blood, and platelets are recruited into the wound site in the inflammatory phase. After the blood clot plugs the wound and the bleeding stops, inflammatory reactions are initiated to clean the wound site. Then, angiogenesis begins, and a provisional connective tissue matrix forms. Vessels and bone-forming cells penetrate the provisional matrix, and immature woven bone is formed in the proliferative phase within 2 weeks after extraction. As this immature woven bone is fragile and non-load-bearing, it is replaced by mature lamellar bone and bone marrow through the remodeling process over several months, although more than 60% of the bone is remodeled within the first 3 months after extraction. After the healing period, most of the original bone volume is decreased, especially on the facial side. A dimensional change is one of the critical factors affecting the clinical outcome of dental implants, and a variety of bone-grafting procedures, including the socket preservation techniques, have been developed to maintain or augment the bone volume during the healing period.

To develop a strategy for improving the amount of new bone formation, understanding wound healing processes at the cellular level is important because growth factors are dynamically orchestrated to recruit the appropriate cells into the defects and stimulate bone formation (Figure 1b) [5]. In inflammatory phases, platelets secrete platelet-derived growth factors (PDGFs) to induce the chemotaxis and proliferation of cells necessary for the wound healing process. Then, pro-inflammatory cytokines—such as interleukin 1 (IL-1), IL-6, and tumor necrosis factor alpha (TNF-α) are secreted—and inflammatory cells migrate into the wound site. In proliferative phases, the angiogenesis process is essential for cellular and nutritional support. Vascular endothelial growth factor (VEGF) and PDGFs regulate angiogenesis, which is closely related to osteogenesis [6]. In addition to vessel formation, osteogenesis occurs in the defects, and BMPs, including BMP-2, play critical roles in the differentiation of osteogenic progenitor cells into osteoblasts and in the mineralization process.

As shown above, growth factors such as VEGF, PDGF, and BMP-2 are related to bone healing and enhance bone regeneration. However, these factors have different actions and different acting times. For example, PDGF is an early-acting growth factor, while BMP-2 is a late-acting growth factor that operates after the initial inflammatory healing process is completed. Therefore, effective delivery systems that consider the action of each growth factor should be designed.

## 3. rhBMP-2 Delivery in Protein Form

Currently available BMP-2 delivery systems use rhBMP-2 in the protein form. To enhance bone regeneration by bone-grafting procedures, rhBMP-2 is applied with bone substitutes or collagen sponges according to the defect morphology. The effects of rhBMP-2 on alveolar bone regeneration have been revealed in various pre-clinical and clinical studies on periodontal regeneration, bone augmentation procedures, and bone reconstruction in peri-implantitis defects [7,8,9,10,11,12,13,14].

Although the amount of native BMP-2 that can be isolated from bone is limited (1–2 μg/kg cortical bone) [15], an innovative method, the recombinant BMP production technology developed by Wozney et al., enables the clinical use of rhBMP-2 [16]. Researchers first succeeded in producing rhBMP-2 genetically, and commercially available rhBMP-2 was then produced via genetic recombination methods using mammalian Chinese hamster ovary (CHO) cells [17]. As the protein synthesis system of CHO cell-derived rhBMP-2 is identical to that in humans, including post-translational modifications, CHO cells produce an active form of N-glycosylated BMP-2. However, the high cost of manufacturing CHO cell-derived rhBMP-2 is one of the reasons hindering the application of rhBMP-2 in clinical practice, even though CHO-derived rhBMP-2 is effective for bone regeneration. Using genetic recombination with *E. coli*, much larger amounts of rhBMP-2 can be produced at low cost; however, *E. coli*-derived rhBMP-2 is not glycosylated and shows reduced biological activity [18]. Nonetheless, recent pre-clinical and clinical studies using *E. coli*-derived rhBMP-2 have shown successful results in bone regeneration in orthopedic and bone augmentation procedures in dentistry [19,20,21].

However, the most concerning issue in rhBMP-2 therapy is the adverse effect related to the dose. As mentioned above, protein growth factors have a common weakness related to short half-lives (minutes to hours) and high clearance rates. Therefore, a high initial dose of rhBMP-2 is applied to defects to maintain an effective in vivo concentration for the healing period because BMP-2 is a late-acting growth factor. In addition, adequate doses can vary according to the carrier system and host conditions. In previous studies, the concentration of rhBMP-2 varied from 0.75 to 2.0 mg/mL [12,22,23]. A supraphysiological dose of rhBMP-2 is related to adverse effects, such as extensive swelling, seroma formation, cystic bone lesion formation and cancer development [24,25]. In addition, Poynton and Lane mentioned concerns related to the use of rhBMP-2, such as the possibility of bony overgrowth, interaction with exposed dura, cancer risk, systemic toxicity, reproductive toxicity, immunogenicity, local toxicity, osteoclastic activation, and effects on distal organs [26].

## 4. BMP-2 Gene Delivery

Gene delivery is an alternative method for transferring growth factors into defect sites. The complementary DNA (cDNA) of human BMP-2 can be transferred via a vector into the site, resulting in the production of BMP-2 in vivo, which induces osteogenic differentiation and mineralization of the site. One of the advantages of BMP-2 gene delivery is the modulation of BMP-2 concentration and duration. Previous studies reported that BMP-2 concentrations in BMP-2 gene delivery applications (100–10,000 pg/mL) are much lower than those in rhBMP-2 applications (0.75–2.0 mg/mL) [27,28]. Depending on the type of vectors carrying the BMP-2 gene, BMP-2 can be released for 2 or 3 weeks at low concentrations. The delivery pattern mimics the action of BMP-2 in the wound healing process, and adverse effects related to high doses of rhBMP-2—such as edema, extensive swelling, implant failure, and immature bone healing—can also be avoided.

Gene delivery is divided into in vivo and ex vivo delivery (Figure 2). In vivo gene delivery directly transfers target genes into the host, either locally or systemically. Ex vivo gene delivery is cell-based gene delivery; cells harvested from the host are transduced with a vector carrying the target genes, and the transduced cells are administered into the defect. Table 1 summarizes the advantages and disadvantages of each delivery method. 

### 4.1. Ex Vivo BMP-2 Gene Delivery

For ex vivo BMP-2 gene delivery, mesenchymal stem cell (MSC)-derived cells, including bone marrow stromal cells (BMSCs), muscle-derived cells, adipose-derived stem cells (ASCs), periodontal ligament stem cells (PDLSCs), and fibroblasts, are frequently used as gene carriers [35,36,37,38,39,40]. Unlike that via in vivo gene delivery, bone regeneration via ex vivo BMP-2 gene delivery produced complete defect closure in calvarial defects within 4 weeks [36,37]. As the efficacy of gene transfer for ex vivo gene therapy is higher than that for in vivo delivery, safety concerns related to the high titers of viral or non-viral vectors used for in vivo delivery can be avoided. However, ex vivo gene delivery methods have limitations related to cell processing and are expensive and time-consuming.

#### 4.1.1. Cells for Ex Vivo BMP-2 Gene Delivery

Although a controversial topic, the cell type influences bone regeneration in ex vivo gene delivery [31,41]. Gafni et al. reported that MSCs are good candidates for gene therapy in bone tissue engineering because MSCs have the potential to differentiate into various lineages, including bone, cartilage, fat, muscle, and ligament [42]. Indeed, most researchers have used MSCs or BMSCs for BMP-2 gene delivery and have shown successful bone regeneration in calvarial defect or mandibular defect models. BMSCs have been widely used for gene delivery and have proven to be effective gene carriers for bone regeneration in ex vivo BMP-2 gene delivery [31,35,43,44,45,46]. Our recent study showed that host conditions at cell harvesting, such as diabetes, affect the osteogenic activity of BMSCs; the BMP-2 secretion pattern after adenoviral BMP-2 gene delivery with cells harvested from diabetic animals is prolonged compared with that after the delivery of cells harvested from healthy subjects (Figure 3) [32]. That is, autologous cells for gene delivery can have impairments stemming from the host, leading to inconsistent treatment outcomes.

Recently, interest in ASCs for BMP-2 gene delivery has been increasing [38,47]. ASCs have advantages, such as easy access and abundant amounts; however, they also have limitations due to low stemness and multipotency [48]. Bougioukli et al. reported that the BMP-2 production and osteogenic differentiation capacity in ASCs were greater than those in BMSCs in in vitro experiments [49]. Vakhshori et al. investigated the cryopreservation of BMP-2-transduced ASCs and revealed that BMP-2 production was limited after the transduced cells were frozen, whereas the BMP-2 production of thawed cells was the same as that of cells that had not been frozen [50].

Dental stem cells isolated from dental follicles, pulp tissue, root apex, and periodontal ligaments exhibit the same characteristics as those of MSCs [51], and PDLSCs have been applied in ex vivo BMP-2 gene delivery [39,52]. Ex vivo BMP-2 gene delivery in dental fields will be discussed in a later section.

In addition to MSCs, fibroblasts and osteoblasts originating from MSCs have also been applied for ex vivo gene delivery. Lee et al. transfected muscle cells with AdBMP-2 and implanted them into calvarial defects. As a result, some of the defects were completely closed after 4 weeks of healing [36,37]. Hirata et al. utilized skin fibroblasts for gene delivery and reported successful bone regeneration in rat calvarial defects [53]. Keeney et al. applied skull-based osteoblasts for BMP-2 gene delivery [54]. Our group reported the possibility of human gingival fibroblasts as a gene vehicle. Gingival fibroblasts are abundant in the oral cavity and are easy to harvest under local anesthesia during dental treatment procedures. Shin et al. implanted human gingival fibroblasts transfected with adenovirus containing BMP-2 into rat calvarial defects and observed complete defect closure after 4 weeks of healing [55].

#### 4.1.2. Viral Vectors for Ex Vivo BMP-2 Gene Delivery

For BMP-2 gene delivery, both viral and non-viral vectors can be utilized regardless of whether the delivery is in vivo or ex vivo (Table 2).

Because viruses such as adenovirus, retrovirus and lentivirus have evolved to transport their gene into host cells efficiently, viral vectors are effective for gene insertion and BMP-2 production. Adenoviral vectors and adeno-associated viral vectors (AAVs) are commonly used in BMP-2 gene delivery [59]. Because members of the retrovirus family, including lentivirus, insert genes into the host chromosome, retroviral vectors are mostly utilized in ex vivo gene delivery. However, retroviruses have concerns related to insertion mutagenesis [33]. Lentiviruses have merits in transfection into non-dividing cells, resulting in the stable production of BMP-2 for long periods of time [50].

Adenoviral vectors are effective for gene transfer because they can infect non-dividing and dividing cells of different types. In addition, as adenoviral vectors are maintained in cells as episomes, BMP-2 production is limited to within a short period of two to three weeks.

However, adenoviruses can cause severe innate and humoral immune responses, which can be dangerous in immunocompromised patients [60]. Moreover, most people already have antibodies to adenoviruses, which can neutralize the effects of gene delivery [57]. AAVs are similar to adenoviruses but are not known to be pathogenic. That is, AAVs are also effective for gene delivery but do not provoke host immune responses. Consequently, AAVs are attractive vectors for gene therapy, and a number of researchers use AAVs for gene therapy, including BMP-2 gene delivery [40,56,61,62].

Most ex vivo BMP-2 gene delivery uses adenoviral vectors and AAVs for gene delivery (Table 3). Lee et al. applied muscle-derived cells transfected by adenoviral vectors containing the BMP-2 gene into mouse calvarial defects and reported that more than 85% of the defects were closed within 2 weeks, with complete closure at 4 weeks [36,37]. Gafni et al. used AAV for BMP-2 gene delivery with tetracycline-sensitive optomotors to regulate bone formation by gene therapy [63]. Hu et al. reported that the lyophilized form of adenoviral vectors was more effective in inducing bone regeneration than was the free form of adenoviral vectors [64]. We established an adenoviral vector system to produce BMP-2 and confirmed that BMP-2 secretion continued until 3 weeks; moreover, bone regeneration by BMP-2 gene delivery was obtained within 4 weeks [33,39,45].

To avoid immune reactions related to adenoviral vectors, Chuang et al. applied a baculovirus for BMP-2 gene delivery [65]. The baculovirus is a kind of insect virus that can enter mammalian cells. The baculovirus does not replicate inside transduced mammalian cells and is non-pathogenic to humans [40]. Baculoviral gene delivery of BMSCs causes transient and mild levels of innate and adaptive immune responses [66,67]. Baculoviral DNA degrades in mammalian cells over time and has an episomal transgene expression pattern, similar to that of adenoviral vectors. Liao et al. observed BMP-2 expression by the baculoviral vector system until 2 weeks and 50% defect closure in the BMP-2 alone group; however, 89% defect closure was observed in the BMP-2/miR-148b group after 12 weeks of healing in a mouse calvarial defect model [47].

Some studies have used lentiviral vectors for ex vivo BMP-2 gene delivery. Generally, since the mechanism of action of BMP-2 is temporary for the wound healing process, the retrovirus family is not suited for BMP-2 gene delivery for the bone defect healing model. Indeed, Blum et al. compared the osteogenic effects of adenoviral vectors and retroviruses for BMP-2 gene delivery, and adenoviral vectors were superior to retroviruses in terms of osteogenic differentiation and bone formation [68]. As a result, lentiviral BMP-2 gene delivery is utilized for the investigation of characteristics related to BMP-2 in bone healing or the development of innovative strategies for BMP-2 gene therapy [49,50,69,70,71]. Alaee et al. suggested dual expression of a suicide gene and BMP-2 gene in a lentiviral vector due to safety concerns [72].

#### 4.1.3. Non-Viral Vectors for Ex Vivo BMP-2 Gene Delivery

In contrast to viral vectors, non-viral vectors are relatively safe with respect to immunogenicity and have a high packaging capacity for manufacturing. However, the poor gene transfection efficiency of non-viral vectors still needs to be improved for further applications. For BMP-2 gene delivery, non-viral vectors—including naked DNA, cationic lipids (liposome-based transfection), cationic polymers (amine-based cationic polymer), and electroporation—have been applied [57].

Among non-viral vectors, cationic polymers, and cationic lipids (liposome-based transfection) have been used for ex vivo BMP-2 gene delivery. Keeney et al. transfected skull-based osteoblasts via plasmid BMP-2 with polymeric dendrimer molecules [54]. BMP-2-producing cells were implanted into mouse critical-sized calvarial defects, and more than 50% of defects were closed within 12 weeks.

Cationic lipids with target genes can be condensed and applied into defects, and Lipofectin and Lipofectamine are commercially available in vitro gene delivery kits [73]. They are composed of a cationic lipid and a helper lipid, such as cholesterol, to promote the condensation of target genes into a stable hexagonal phase structure for improving transfection efficiency [74]. For BMP-2 gene delivery in the oral and facial regions, liposome-mediated BMP-2 genes are transferred into cells, such as BMSCs or MSCs, which were applied to the defect sites. Blum et al. attempted liposome-mediated BMP-2 gene delivery for bone regeneration in rat calvarial defects, and approximately 30% of the defects were closed after 30 days [67]. Park et al. also conducted experiments comparing adenoviral vectors and liposomes for BMP-2 gene delivery [35]. In their study, the BMSCs were transfected by adenoviral vectors or liposomes carrying the BMP-2 gene, and the gene-delivered cells were applied in rat mandibular defects. As a result, complete defect closure was observed in 6 weeks in the liposome group and 4 weeks in the adenoviral group. Tang et al. applied BMSCs transfected with plasmid BMP-2/liposomes into an osteoporotic rat mandibular defect model and observed newly formed bone at 4 weeks and mature healing at 8 weeks [75].

#### 4.1.4. Bone Regeneration via Ex Vivo BMP-2 Gene Delivery

Gene-delivered cells are carried by a hydrogel into the defect with or without a bone substitute. BMP-2-producing cells are effective for bone regeneration, but space maintenance for new bone formation is also critical in bone regeneration, especially in large defect healing.

Regarding animal experimental models, most of the studies were tested in a critical-sized calvarial defect model. Gelatin or collagen sponges were utilized for scaffolds and were found to be successful for bone regeneration [35,36,37,45,52,53,54,68,76]. Polymers alone or polymers with inorganic minerals were also applied in the defects [44,46,47,55]. However, the amount of bone regeneration was influenced by differences in the vector system, not by scaffolds.

In an experimental model related to the dental field, direct injection of an aqueous solution containing genetic materials was applied in a mandibular distraction model. However, the statistical significance between BMP-2 gene delivery and BMP-2 protein delivery was not observed within this delivery system [77]. In maxillary sinus graft models and dental implant-related models, bone substitutes such as beta-tricalcium phosphate or deproteinized bovine bone minerals were used to maintain the space for bone regeneration during healing periods, similar to conventional bone regeneration procedures in clinical settings [38,39,78,79]. Bone regeneration was also significantly enhanced in studies using a viral vector system, and the results were consistent with those obtained from calvarial experimental models.

#### 4.1.5. Dental Application via Ex Vivo BMP-2 Gene Delivery

In the dental field, PDLSCs are involved in MSCs and differentiate into osteoblasts, cementoblasts, and fibroblasts and generate periodontium supporting natural teeth, which consists of bone, cementum, periodontal ligaments, and connective tissues. Since we previously confirmed that BMSCs and PDLSCs induce alveolar bone regeneration at a similar level [80], we applied PDLSCs for ex vivo BMP-2 gene delivery to restore the defects induced by peri-implantitis [39].

Peri-implantitis is an inflammatory disease related to bacterial accumulation around a dental implant and results in bone loss around the implants, leading to implant failure [81]. Peri-implantitis has similar pathophysiological characteristics except for its progressive and destructive features. In addition, peri-implantitis defects are challenging for bone regeneration above the defects as well as re-osseointegration, which means direct contact of the bone to the implant and is a key factor in bearing the occlusal force in the oral cavity. As such, innovative approaches are necessary for bone regeneration of peri-implantitis defects. For this purpose, we applied PDLSCs transfected with AdBMP-2 (BMP-2/PDLSCs) for delivery to peri-implantitis defects, which were experimentally induced and where the level of alveolar bone loss was half that of the implant [39]. After 3 months of healing, the BMP-2/PDLSCs produced significantly greater amounts of newly formed bone above the bottom of the defects and resulted in re-osseointegration over the implant (Figure 4). In addition, lace-like immature woven bone, which is frequently observed in rhBMP-2 application cases [25], was not observed in the BMP-2/PDLSC groups. Furthermore, the maturity of the newly formed bone was similar to that of the old bone. This study was consistent with Yi et al.’s study, which compared the rhBMP-2 protein with PDLSCs vs BMP-2-producing PDLSCs and revealed the superiority of BMP-2-producing PDLSCs in bone regeneration in calvarial defects [52].

### 4.2. In Vivo BMP-2 Gene Delivery

For in vivo gene delivery, direct injection of genetic materials can be applied. However, Zhou et al. reported that directly targeted gene materials do not produce the desired effect due to a rapid clearance rate, rapid enzymatic degradation, nonspecific biodistribution, and low cellular uptake [85]. Accordingly, viral or non-viral vectors are utilized to protect genetic materials and transfer them to cells or defect sites. However, in vivo gene delivery raises concerns related to the direct injection of excessive amounts of adenovirus needed to induce bone regeneration, leading to excessive immune reactions [86]. Recently, plasmids carrying the BMP-2 gene have been increasingly used for gene delivery and have been applied to defects with bone substitutes or polymers for bone regeneration in oral and facial regions.

#### 4.2.1. Viral Vectors for In Vivo BMP-2 Gene Delivery

Similar to their roles in ex vivo gene delivery, adenovirus, and AAV are also commonly used viral vectors in in vivo BMP-2 gene delivery. Alden et al. confirmed significant bone healing in rat mandible defects via the direct injection of adenoviral vectors containing BMP-2 or BMP-9 genes [87]. Ashinoff et al. also applied local injection into the mandibular distraction osteogenesis site [88].

Ben Arav et al. coated bone allograft materials with freeze-dried recombinant AAV vector encoding the BMP-2 gene and implanted them into mouse calvarial defects [89]. Because the single-stranded AAV did not produce superior results compared to those produced by the uncoated allografts or autografts, the authors utilized self-complementary AAV, which was expected to increase the transfection efficiency up to 6-fold [90]. As a result, complete closure of defects was achieved within 4 weeks after implantation.

#### 4.2.2. Non-Viral Vectors for In Vivo BMP-2 Gene Delivery

For in vivo BMP-2 gene delivery, BMP-2 genes are transferred into defects either via naked plasmid DNA with or without electroporation or via plasmid DNA coated with a polymer. Because liposomes have not been used as vehicles for BMP-2 gene delivery in in vivo gene delivery, the use of liposomes as non-viral vectors will be discussed in a later section on ex vivo BMP-2 gene delivery.

Electroporation is a technology that increases cell permeability by applying short, high-voltage pulses to cells to introduce target genes into the cells [91]. Aslan et al. performed BMP-2 gene delivery through electroporation into human MSCs, which produced BMP-2 protein for 14 days [92]. In addition, they injected cells carrying BMP-2 into the muscle, resulting in ectopic bone formation. Wu et al. utilized electroporation for BMP-2/VEGF, BMP-2, or VEGF gene transfer into the defect sites of a rabbit mandibular distraction osteogenesis model [93]. As a result, both BMP-2 and BMP-2/VEGF led to a significant amount of new bone formation, although the BMP-2/VEGF combined group showed significant bone regeneration compared to that of the BMP-2 alone group. Recently, Kawai et al. transferred the BMP-2/BMP7 gene into the periodontal tissue of maxillary first molars of rats via electroporation and confirmed the expression of BMP-2/BMP7 in rat periodontal tissues and new bone formation around the original alveolar bone of teeth [94]. In addition to using BMP-2, Tsuchiya et al. tried BMP4 gene delivery for periodontal bone regeneration via electroporation but failed to produce significant bone formation [95].

Cationic polymers, such as polyethyleneimine (PEI), chitosan, and polylactic-co-glycolic acid (PLGA), have been applied for BMP-2 gene delivery. Most polymers are cationic and are simply complexed to nucleic acids via electrostatic interactions. Due to this convenience in use, polymers are widely studied materials in non-viral gene delivery [96]. In BMP-2 gene delivery, Chew et al. confirmed that triacrylate/amine cationic polymers are advantageous in slowing the degradation rate of naked DNA. Chitosan is a natural polymer but is not strong enough to deliver complex gene materials [97]. In a BMP-2 gene delivery study, a chitosan-based thermosensitive hydrogel containing a plasmid BMP-2 gene was injected into rat calvarial defects and enhanced new bone formation [98]. However, complete defect closure was not observed after 4 weeks of healing, and most defects were filled with bone after 8 weeks. The most widely applied synthetic cationic polymer for gene delivery, PEI is known to have a proton sponge effect, which is advantageous for escaping from endolysosomes into the cytoplasm [99]. PLGA is a clinically approved biomaterial that degrades into non-toxic molecules. However, due to the negative surface charge of PLGA, PLGA alone cannot complex with nucleic acid. Instead, PLGA is used as a nanoparticle with cationic PEI for electrostatic loading. Qiao et al. encapsulated PEI nanoparticles with a plasmid BMP-2 gene by using PLGA and applied these nanoparticles to rat calvarial defects [100]. However, this complex gene delivery system did not result in complete defect closure.

#### 4.2.3. Bone Regeneration through In Vivo BMP-2 Gene Delivery

For bone regeneration, scaffolds delivering vectors carrying genetic materials into defect sites are a critical factor affecting bone regeneration by BMP-2 gene delivery (Table 4). Viral or non-viral vectors carrying the BMP-2 gene have been directly injected into defect sites, producing significant bone formation [87,88,101]. In other studies, vectors have been delivered into sites with hydroxyapatite [28,89], polymers, such as PLGA [102], or polymer (poly(d,l-lactide))-coated titanium discs [103]. However, a variety of scaffolds were applied in most of the studies, and a conclusion could not be reached regarding the superiority of the scaffolds in in vivo BMP-2 gene delivery.

Animal models of critical-sized calvarial defect are most frequently used to test the efficacy of in vivo BMP-2 gene delivery. In most of the studies using non-viral vectors for in vivo BMP-2 gene delivery, complete defect closure in a critical-sized calvarial defect was not observed, while more than 80% defect closure was observed at 8 weeks when the researchers utilized adenoviral vectors carrying the BMP-2 gene (AdBMP-2) with a PLGA nanofibrous scaffold; approximately 20% defect closure was observed at 4 weeks of healing. In a study using rat mandibular ramus defect models, Alden et al. reported that complete defect closure was observed 12 weeks after direct injection of AdBMP-2 [87]. Similarly, Kolk et al. formed mandibular ramus defects and covered them with a titanium disc coated with a PDLLA polymer that embedded the plasmid BMP-2 gene [103]. As a result, new bone formation was effective, and complete defect closure was observed after 16 weeks. However, the amounts of newly formed bone showed an inverse dose dependency. In a study using mandibular distraction osteogenesis models, Ashinoff et al. applied AdBMP-2 to the defects and produced significant amounts of bone formation after 4 weeks, while the control showed a similar level of bone healing [100]. In a study using periodontal defect models, Li et al. produced significant amounts of new bone formation with PDL 8 weeks after the injection of a chitosan-based thermosensitive hydrogel scaffold with plasmid BMP-2 [98]. BMP-2/BMP7 gene transfer also led to an increase in the mineral apposition rate of alveolar bone compared to that in the control group [94].

### 4.3. Complementary Strategies of In Vivo and Ex Vivo BMP-2 Gene Delivery

In recent years, muscle grafts or adipose tissues have been employed to deliver genetic materials (gene-activated matrices) because the former are abundant sources of MSCs and are easy to harvest [105,106]. In addition, these methods are advantageous because gene delivery to tissues can prevent the complicated ex vivo cell expansion processes necessary for ex vivo gene therapy. Liu et al. implanted muscle tissue transfected by AdBMP-2 [104]. As a result, BMP-2 gene-activated muscle grafts were effective in bone regeneration in rat calvarial defects. Ren et al. attempted the transfection of adipose tissue fragments harvested from rat subcutaneous fat by using AdBMP-2 and confirmed that the cells in adipose tissues differentiated into osteoblasts [107]. In addition, Virk et al. proposed a “same-day” strategy for ex vivo BMP-2 gene delivery, although related to orthopedic research [108]. The researchers utilized a buffy-coat layer from bone marrow for cells carrying the BMP-2 gene and applied the layer to the femoral fracture defect, achieving successful union and bone volume.

### 4.4. Combined Approaches Improving Bone Regeneration

Since both osteogenesis and angiogenesis are critical for bone regeneration, combinational strategies are also applied to BMP-2 gene delivery. He et al. transfected MSCs and endothelial progenitor cells (EPCs) with adenoviral vectors containing BMP-2 [82]. As a result, new bone formation and vascular formation were increased in rat calvarial defect models. In other studies, BMP-2 combined with VEGF has been applied to enhance bone regeneration. As BMP-2 and VEGF act throughout the proliferation and modelling/remodeling phases, they were delivered as BMP-2/VEGF genes, and synergistic effects compared to those in the BMP-2 alone group were reported [46,99,103].

To improve bone regeneration, we designed a strategic delivery system according to the properties of each growth factor. Wound healing processes are orchestrated by various growth factors and cytokines. In addition, various molecules are coordinated with spatial and temporal patterns [109]. Accordingly, BMP-2 gene therapy can be applied as a component of combinational therapy for bone regeneration. While PDGF-BB is an early-acting growth factor that recruits cells necessary for wound healing and induces cell proliferation, BMP-2 is a late-acting molecule that induces bone regeneration in the defect after the initial healing process. Accordingly, we applied the PDGF-BB protein for short-term action and the BMP-2 gene for sustained BMP-2 action. We concluded that dual delivery, considering its mechanism of action, was effective for bone regeneration in terms of quantity and quality [45].

## 5. Conclusions

BMP-2 gene delivery is one option for achieving bone regeneration in patients with impaired bone healing, such as patients with diabetes or those with an irradiated site. In dentistry, bone-grafting procedures are closely connected with the ability of dental implant treatments to restore oral functions. One of the advantages of BMP-2 gene therapy is the regeneration of bone with sufficient quality to bear loads during daily function due to the prolonged secretion of BMP-2 at low concentrations. In experimental models designed to prove the effect of BMP-2 gene delivery, critical-sized calvarial defects were used as a standard for screening tests. Recently, pre-clinical translation of BMP-2 gene therapy has begun, and maxillary sinus-related and dental implant-related models have been utilized. Periodontal regeneration via BMP-2 gene delivery has also been attempted, although a small animal model was used. Considering the bone healing process, adenoviral vectors or AAVs are preferred for oral and maxillofacial regions if safety concerns can be addressed. In terms of stable gene transfer, BMP-2 production and bone regeneration, ex vivo BMP-2 gene delivery using autologous cells has some merits. However, the high cost of good manufacturing practices (GMPs) related to ex vivo cell culture systems increases the difficulty of this clinical application. Recently, commercially available gene therapy has begun to develop in certain areas, including orthopedics and dentistry. For rheumatoid arthritis, a clinical trial is in progress in the Netherlands; this study involves AAV carrying the interferon-beta gene [110]. In the maxillofacial region, “Nucleostim”, consisting of octacalcium phosphate and plasmid DNA encoding VEGF, is being tested in ongoing phase III clinical trials [111]. To achieve BMP-2 gene-activated bone regeneration in the oral and maxillofacial regions, efficient vector systems with reduced immunogenicity and a stable cell supply, such as the umbilical cord, gingival tissue, and adipose tissue (either autologous or allogenic), are critical factors for gene delivery.

## Figures and Tables

**Figure 1 pharmaceutics-11-00393-f001:**
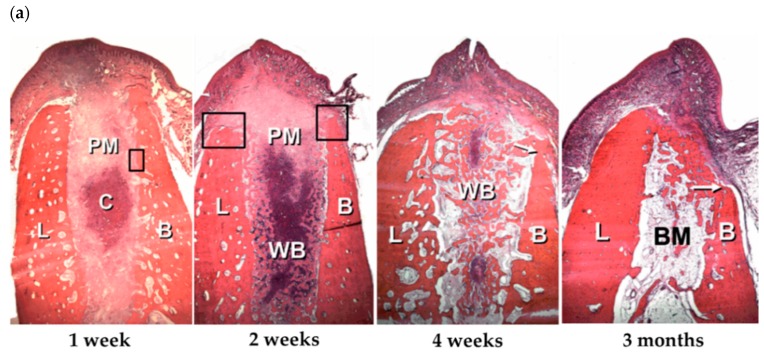
(**a**) Extraction socket-healing process. (BC, blood clot; B, buccal; L, lingual; PM, provisional matrix. WB, woven bone; BM, bone marrow; H&E staining; original magnification 16×). Reproduced with permission from Araujo et al. [4]. Copyright © 2005. John Wiley and Sons. (**b**) Growth factors related to bone wound healing. (PDGF, platelet-derived growth factor; VEGF, Vascular endothelial growth factor; BMPs, bone morphogenetic proteins; TGF-β, Transforming growth factor-beta) Reproduced with permission from Hollinger et al. [5]. Copyright © 2008. Wolters Kluwer Health, Inc.

**Figure 2 pharmaceutics-11-00393-f002:**
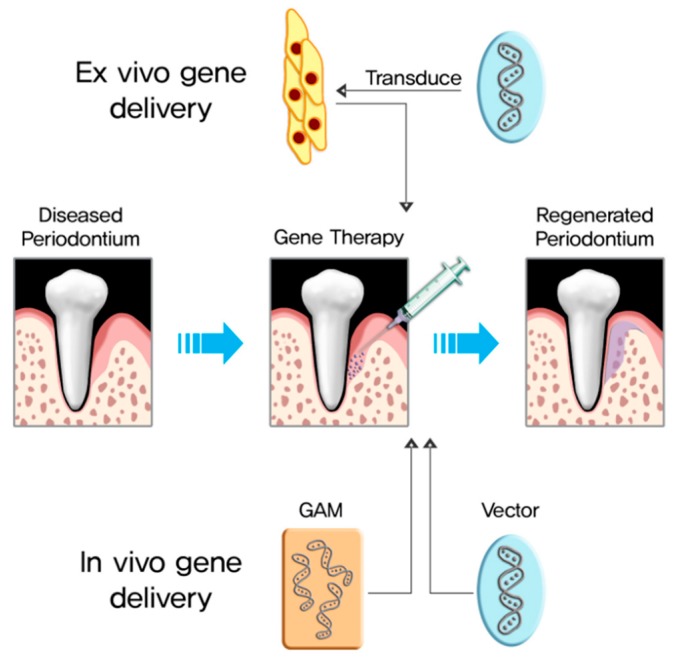
Regeneration strategy for the reconstruction of periodontal tissue through gene therapy.

**Figure 3 pharmaceutics-11-00393-f003:**
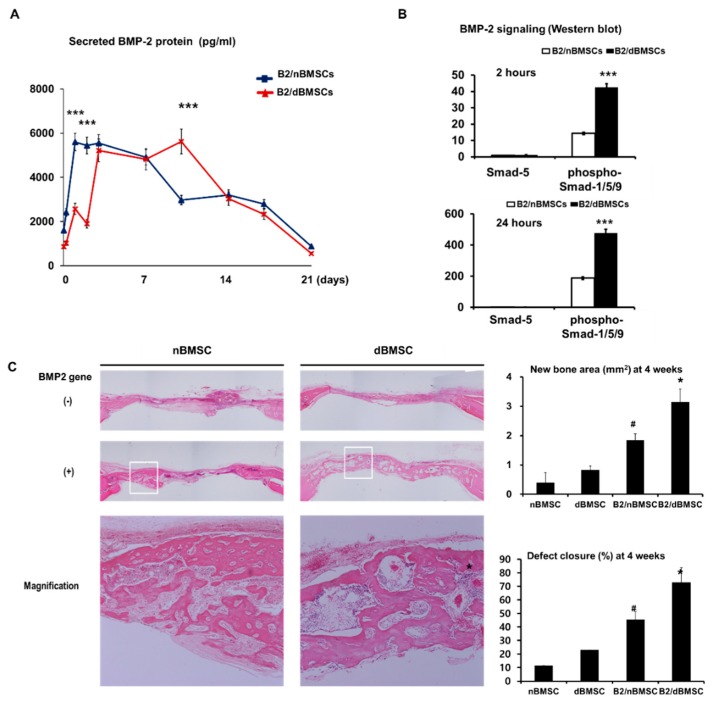
Effect of diabetes on BMP-2 production and bone regeneration of BMSCs. (**A**) BMP-2 secretion of AdBMP-2-transfected nBMSCs vs that of AdBMP-2-transfected dBMSCs. (‘‘*’’indicates a significant difference, with * *p* < 0.05, ** *p* < 0.01, and *** *p* < 0.001 (paired t-test)) (**B**) BMP-2 signaling pathway analyzed by western blotting. (‘‘*’’indicates a significant difference, with * *p* < 0.05, ** *p* < 0.01, and *** *p* < 0.001 (paired t-test)) (**C**) Bone regeneration by AdBMP-2-transfected nBMSCs vs that by AdBMP-2-transfected dBMSCs. (‘‘*’’ indicates a significant difference from the other groups (ANOVA with Tukey’s post hoc test, *p* < 0.001) and ‘‘#’’ indicates a significant difference from the other groups), non-diabetic bone marrow stromal cells, nBMSC; AdBMP-2-transfected nBMSCs, B2/nBMSCs; diabetic BMSCs, dBMSCs; AdBMP-2-transfected dBMSCs; B2/dBMSCs. Reproduced from Park et al. [32]. Copyright © 2018. Mary Ann Liebert Inc.

**Figure 4 pharmaceutics-11-00393-f004:**
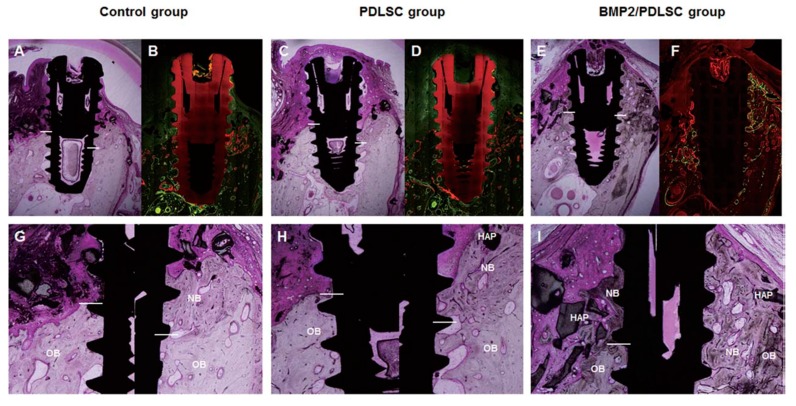
Bone regeneration and re-osseointegration around peri-implantitis defects by AdBMP-2/PDLSCs. Upper pannel: light microscopic (**A**,**C**,**E**) and confocal laser scanning microscopic (**B**,**D**,**F**) photographs of histological sections of each experimental group at 12 weeks (original magnification × 1.5). Lower panel: magnification of newly formed bone and re-osseointegration within defects (**G**,**H**,**I**; original magnification × 4), NB, new bone; HAP, hydroxyapatite particles; OB, old bone; bone labeling: 4 weeks (green), 8 weeks (orange); multiple stains and undecalcified ground sections; bar indicates bottom of defect. Reproduced with permission from Park et al. [39]. Copyright © 2014 Wiley Periodicals, Inc.

**Table 1 pharmaceutics-11-00393-t001:** Advantages and disadvantages of ex vivo and in vivo BMP-2 gene delivery

Delivery Type	Advantages	Disadvantages
ex vivo	Gene transfer is limited to the target cell population and not to other cells or tissues	Expensive and time-consuming process
	Can use gene transfer to genetically modify stem cells, e.g., embryonic stem cells and iPSCs [29,30]	Complicated manipulation including cell harvesting, cell expansion and transfection
	High efficacy	The outcome can be influenced by the carrier cells [31,32]
	Low quantity of vectors is necessary for desired therapeutic effects	
	Minimal immune recognition of the gene vectors [33]	
in vivo	Simple process via direct injection into the site or intravenous administration	Low efficacy
	Avoids complicated process related to cells	High quantity of vectors is necessary for desired therapeutic effects
	Relatively low cost	Induction of immune reaction due to direct exposure of vectors
		Difficult to target the cell population of interest
		Vector system is potentially toxic [34]

**Table 2 pharmaceutics-11-00393-t002:** Advantages and disadvantages of viral and non-viral vectors in gene delivery.

Type of Vectors	Advantages	Disadvantages
Viral vectors	High gene transduction efficiency	Difficult to manufacture, produced in low virus titers
	Transgene expression can be controlled by virus (transient expression or persistent expression)	Immune reactions to virus [56]
	Can target specific cell types such as dividing cells or non-dividing cells [33,57]	Limitation in packaging capacity e.g., 4.5 kb for AAV vectors [33,57]
		Safety concerns e.g., insertion mutagenesis [58]
Non-viral vectors	Simple manufacturing	Low in vivo gene transduction efficiency [57]
	Low cost	High quantity for therapeutic effects
	Low immunogenicity	Cannot target specific cell types
	High packaging capacity	Toxicity related to materials [34]

**Table 3 pharmaceutics-11-00393-t003:** Ex vivo BMP-2 gene therapy for bone regeneration in oral and facial regions.

References	Cells	Vectors	Transgene	Carrier	Model	Results
Lee et al. 2001,2002 [36,37]	Muscle-derived cells	Adenovirus	BMP-2	Collagen sponge	Mouse calvarial defect	Mouse calvarial defects treated with BMP-2-producing muscle cells had >85% closure within two weeks and 95–100% closure within four weeks.
Blum et al. 2003 [68]	MSCs	Adenovirus Retrovirus Cationic lipid	BMP-2	Titanium mesh scaffold	Rat calvarial defect	All viral and non-viral vectors carrying the BMP-2 gene were effective in bone regeneration. However, adenoviral vectors resulted in slightly significantly increased amounts of newly formed bone compared to those achieved with other vectors and the control group.
Hirata et al. 2003 [51]	Skin fibroblasts	Adenovirus	BMP-2 or Runx2	PDLLGA /gelatin sponge	Rat calvarial defect	AdBMP-2-transplanted skin fibroblasts were effective on new bone formation. However, cells with AdRunx2 were insufficient in inducing bone repair.
Park et al. 2003 [35]	BMSCs	Adenovirus Liposome	BMP-2	Collagen sponge	Rat mandibular defect	Both liposome-mediated and adenoviral BMP-2 gene transfer to BMSCs successfully achieved the healing of critical-size bone defects in rats.
Gafni et al. 2004 [63]	MSCs	AAV	BMP-2	Collagen sponge	Mouse calvarial defect	AAV-BMP-2 with a tetracycline-sensitive promotor was effective in regulation of bone formation by gene therapy.
Hu et al. 2007 [40]	Fibroblasts	Adenovirus	BMP-2	Gelatin sponge, HA disc	Rat calvarial defect	Lyophilized AdBMP-2 in a gelatin sponge was more effective than the free form of adBMP-2 in rat calvarial defects.
Koh et al. 2008 [76]	Fibroblasts	Adenovirus	BMP-2/7	Gelatin sponge	Mouse calvarial defect	AdBMP-2/7-transduced cells were more effective in healing cranial defects than were cells individually transduced with AdBMP-2 or BMP7.
Tang et al. 2008 [74]	BMSCs	Liposome/Plasmid	BMP-2	Coral hydroxyapatite matrix	Rat mandibular defect osteoporotic model	Autogenous cells transfected with pBMP-2 promoted bone formation in osteoporotic rats.
Steinhardt et al. 2008 [43]	BMSCs	Adenovirus	BMP-2	Collagen sponge	Mouse mandibular defect	Application of genetically engineered BMP-2-producing BMSCs into a mandibular defect led to tissue regeneration at the defect site.
Wang et al. 2009 [69]	Skin fibroblasts	Retrovirus	BMP-2	Gelatin sponge	Rat calvarial defect	Autologous BMP-2-modified skin fibroblasts successfully led to bone regeneration in rat calvarial defects. Fibroblasts could be effectively used in ex vivo gene therapy for local bone repair.
Chang et al. 2009 [44]	BMSCs	Adenovirus	BMP-2	Gelatin/ tricalcium phosphate ceramic/ glutaraldehyde biopolymer	Rat calvarial defect	AdBMP-2-transfected cells with the gelatin/tricalcium phosphate ceramic/glutaraldehyde biopolymer strongly enhanced the bone healing of critical-size bicortical craniofacial defects.
Shin et al. 2010 [55]	Human gingival fibroblasts (HGF)	Adenovirus	BMP-2	Collagen matrix	Rat calvarial defect	AdBMP-2-transfected HGF promoted osseous healing of calvarial defects compared with that achieved in the other groups.
Chuang et al. 2010 [64]	Human MSCs	Baculovirus	BMP-2	PLGA scaffolds	Rat calvarial defect	Although a baculovirus was effective for BMP-2 gene transfer into cells, the use of hMSCs could not overcome the immunological barrier in rats.
Kroczek et al. 2010 [77]	BMSCs	Plasmid-liposome	BMP-2	Direct injection with an aqueous solution of osteoinductive substances	Minipig distraction osteogenesis	BMP-2 expression was maximal in the pBMP-2 group although bone regeneration was not significantly enhanced in the pBMP-2 group compared to that in the rhBMP-2 and rhBMP-7 groups.
Xia et al. 2011 [78]	BMSCs	Adenovirus	BMP-2 Nell-1	Beta-tricalcium phosphate	Rabbit maxillary sinus graft	BMP-2 and Nell-1 genes showed a synergistic effect on osteogenic differentiation of BMSCs and promoted new bone formation and maturation in a rabbit maxillary sinus model.
Lin et al. 2012 [46]	BMSCs	Baculovirus	BMP-2 VEGF	Disc-shaped PLGA scaffolds	Rabbit calvarial defect	Baculoviral vectors were effective in BMSCs for sustained BMP-2/VEGF expression and the repair of critical-size calvarial defects.
He et al. 2013 [82]	MSCs EPCs	Adenovirus	BMP-2	Injectable and porous nano calcium sulfate/alginate	Rat calvarial defect	The combination of BMP-2 gene-modified MSCs and EPCs in injectable scaffolds increased new bone and vascular formation.
Park et al. 2013 [45]	BMSCs	Adenovirus	BMP-2	Collagen gel	Rat calvarial defect	Dual delivery of autologous AdBMP-2-transfected BMSCs and rhPDGF-BB enhanced both the quality and quantity of new bone formation.
Jhin et al. 2013 [79]	BMSCs	Adenovirus	BMP-2	Deproteinized bovine bone mineral	Rabbit maxillary sinus Dental implant placement	BMSCs with AdBMP-2 transfection resulted in earlier bone healing with increased amounts in the maxillary sinus defects when dental implants were simultaneously placed.
Jin et al. 2014 [83]	BMSCs	PEI-alginate/Plasmid	BMP-2	Cell sheet	Rat calvarial defect	PEI-al nanocomposites as a carrier for pBMP-2 gene delivery to BMSCs was effective. Bone regeneration was slightly enhanced by BMP-2- producing BMSCs compared to that in the control group.
Liao et al. 2014 [47]	ASCs	Baculovirus	BMP-2/miR-148b	Disc-shaped poly (L-lactide-co-glycolide) (PLGA) scaffolds	Mouse calvarial defect	Co-transduction of hASCs with BMP-2/miR-148b via baculovirus vectors enhanced and prolonged BMP-2 expression and synergistically promoted bone regeneration.
Park et al. 2015 [39]	PDLSCs	Adenovirus	BMP-2	HA particle with collagen gel	Beagle peri-implantitis defect	PDLSCs transfected by AdBMP-2 produced significantly greater amounts of new bone in peri-implantitis defects than those produced in other groups.
Keeney et al. 2016 [54]	Skull-derived osteoblasts	Cationic amine polymer/Plasmid	BMP-2	PLGA	Mouse calvarial defect	Skull-derived osteoblasts transfected by pBMP-2 led to substantially accelerated bone repair as early as two weeks, which continued to progress over 12 weeks.
Yi et al. 2016 [52]	Human PDLSCs	Adenovirus	BMP-2	Block-type biphasic calcium phosphate	Rat calvarial defect	hPDLSCs showed an inhibitory action on BMP-2-induced osteogenic differentiation. hPDLSCs-transfected AdBMP-2 produced lower amounts of newly formed bone than did hPDLSCs with rhBMP-2 protein.
Xu et al. 2016 [38]	ASCs	Adenovirus	BMP-2	Beta-tricalcium phosphate	Beagle peri-implantitis defect	ASCs transfected by adenoviral BMP-2 produced significant amounts of new bone formation and re-osseointegration compared to those in control groups.
Vural et al. 2017 [84]	BMSCs	Liposome/Plasmid	BMP-2	Gelatin sponge	Rat calvarial defect	pBMP-2 gene delivery in BMSCs effectively led to bone regeneration in rat calvarial defects.
Park et al. 2018 [32]	BMSCs	Adenovirus	BMP-2	Collagen gel	Rat calvarial defect Diabetic model	In diabetic animals, BMP-2 gene therapy using diabetic cells was more effective in new bone formation than was BMP-2 gene therapy using non-diabetic cells.

**Table 4 pharmaceutics-11-00393-t004:** In vivo BMP-2 gene therapy for bone regeneration in oral and facial regions.

References	Administration	Vector	Transgene	Model	Results
Alden et al. 2000 [87]	Direct injection	Adenovirus	BMP-2 BMP9	Rat mandible defect	Significant bone healing was observed in the BMP gene transfer group.
Ashinoff et al. 2004 [100]	Direct injection	Adenovirus	BMP-2	Rat distraction osteogenesis	Local injection of AdBMP-2 increased bone regeneration during distraction osteogenesis.
Chew et al. 2011 [97]	Gelatin microparticle	Triacrylate/ amine polymer/ Plasmid	BMP-2	Rat calvarial defect	Triacrylate/amine gelatin effectively slowed the degradation rate compared to that of naked pDNA.
Zhang et al. 2011 [28]	Fibronectin/apatite	Plasmid	BMP-2	Rat calvarial defect	Bone formation in the pBMP-2 with fibronectin/hydroxyapatite group was enhanced compared to that in the control group.
Wu et al. 2012 [93]	Electroporation	Plasmid	BMP-2 BMP-2/ VEGF	Rabbit distraction osteogenesis	pBMP-2/VEGF gene transfer with electroporation was effective for bone regeneration relative to the control.
Liu et al. 2012 [104]	Muscle tissue	Adenovirus	BMP-2	Rat calvarial defect	The amount of new bone in muscle tissue transduced with AdBMP-2 was more than twice that in the control.
Ben Arav et al. 2012 [89]	Bone allograft	AAV	BMP-2	Mouse calvarial defect	Self-complementary-rAAV-BMP-2-coated allografts were more effective for bone regeneration than were single strand-rAAV-BMP-2-coated allografts, the effects of which were not significantly different from those of autografts or uncoated allografts.
Qiao et al. 2013 [100]	PLGA nanoparticle	PEI nanoparticles encapsulated by PLGA/plasmid	BMP-2	Rat calvarial defect	pBMP-2 gene delivery using a PLGA nanoparticle delivery system was effective for producing BMP-2 cDNA and new bone formation.
Kolk et al. 2016 [102]	Poly(d, l-lactide) (PDLLA)-coated titanium disc	PEI/plasmid	BMP-2	Rat mandibular defect	pBMP-2 gene delivery using a copolymer was successful for controlling new bone formation with an inverse dose dependency.
Xie et al. 2017 [88]	Direct injection	Plasmid	BMP-2/VEGF	Rabbit distraction osteogenesis	The direct injection of pBMP-2/VEGF promoted bone formation in the distraction gap with the upregulation of TGF-β1 expression.
Li et al. 2017 [98]	Injectable thermosensitive hydrogel scaffold	Chitosan/plasmid	BMP-2	Rat calvarial defect Dog mandibular defect	An injectable chitosan-based thermosensitive hydrogel scaffold (CS/CSn-GP) enhanced new bone formation in rat calvarial defects and bony defect healing in beagle dogs.
Zhu et al. 2017 [103]	Electrospun PLGA nanofibrous scaffold	Adenovirus	BMP-2	Rat calvarial defect	A lyophilized PLGA nanofibrous scaffold efficiently released functional AdBMP-2 to transduce local cells, resulting in hBMP-2 secretion and promoting new bone formation in vivo.
Kawai et al. 2017 [94]	Electroporation	Plasmid	BMP-2/7	Rat periodontal tissue	The mineral apposition rate of the alveolar bone following BMP-2/7 gene transfer was significantly higher than that in the control group.

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
