# Peer review of "BMP-2 Gene Delivery-Based Bone Regeneration in Dentistry"

_pharmaceutics, 2019, doi:10.3390/pharmaceutics11080393_

Round 1
Reviewer 1 Report
Review article " BMP2 Gene Delivery-Based Bone Regeneration in Dentistry " is very interesting and significant for researchers and clinicians in Dentistry. I would like to suggest you to consider several points before the article would be published.
Point 1
They should reconsider the section " 1 introduction ". This section seems not to be necessary in this review. If they could delete or simplify his section, it could be clear that they would like to focus on gene therapy in this review.
Point 2
They should discuss the comparison with BMP-2 gene therapy and other BMP family in the section of 4.1.2 Non-viral vectors for in vivo gene therapy. Because Tsuchiya et al, 2016 also tried to delivery BMP-4 gene to periodontal tissues of rats, they could not find clear new alveolar bone formation in the original bone tissues.
Point 3
They should add references for line 227, 228, 231.
Author Response
Point 1
They should reconsider the section " 1 introduction ". This section seems not to be necessary in this review. If they could delete or simplify his section, it could be clear that they would like to focus on gene therapy in this review.
Thank you for your comments. As per your recommendation, we have revised the introduction to clarify the aim of this review.
Point 2
They should discuss the comparison with BMP-2 gene therapy and other BMP family in the section of 4.1.2 Non-viral vectors for in vivo gene therapy. Because Tsuchiya et al, 2016 also tried to delivery BMP-4 gene to periodontal tissues of rats, they could not find clear new alveolar bone formation in the original bone tissues.
As per your comments, we have added the discussion about Tsuchiya et al.’s study in the paragraph.
Point 3
They should add references for line 227, 228, 231.
Thank you for your comments. We have added the references to the sentences.
Reviewer 2 Report
This review summarizes current advances in bone regeneration via BMP-2 delivery with a focus on gene therapy in dentistry. Overall, the manuscript is well-written and well-organized. There are only few suggestions and/or comments. Specifically, since the review discusses different genetic techniques and approaches, additional tables should be included to summarize their advantages and disadvantages (Comments 1 and 2) in order to provide readers with a clearer picture of various methods currently available for gene therapy/delivery.
1. Please create a table that discusses advantages and disadvantages of in-vivo versus ex-vivo (BMP-2) gene delivery approaches as well as potential improvements for each.
2. Please create a table that summarizes advantages and disadvantages of different techniques (virus versus non-viral vectors) used for gene delivery/therapy.
3. For comments 1 and 2, transfection efficiency should also be discussed.
4. Fig. 2a: Abbreviations (B, BM, C, L, PM, WB) should be defined in the figure legends.
5. Line 157: Figure 3 related to rhBMP-2 was not included in the manuscript.
6. Lines 217-218 about lentivirus: References should to be cited. In addition, lentivirus or retrovirus has recently been widely used in the fields, and thus the authors may want to expand the discussion (lines 375-382) further.
7. Lines 226-231: Examples related to the use of adeno-associated viral vectors (AAVs) for in-vivo gene therapy should be discussed and cited.
8. Lines 305-307 about the use of each of different cell types: References should to be cited.
9. Fig. 4: Abbreviations (nBMSCs vs dBMSCs) should be defined in the figure legends.
10. Minor language editing and proofreading is needed. For example, line 44 – it should be “less functional” instead of “malfunction”.
11. Please review the reference list thoroughly; for example, reference #63 (line 710).
Author Response
This review summarizes current advances in bone regeneration via BMP-2 delivery with a focus on gene therapy in dentistry. Overall, the manuscript is well-written and well-organized. There are only few suggestions and/or comments. Specifically, since the review discusses different genetic techniques and approaches, additional tables should be included to summarize their advantages and disadvantages (Comments 1 and 2) in order to provide readers with a clearer picture of various methods currently available for gene therapy/delivery.
1. Please create a table that discusses advantages and disadvantages of in vivo versus ex vivo (BMP-2) gene delivery approaches as well as potential improvements for each.
- Thank you for your comments. As per your recommendation, we have added a table showing the advantages and disadvantages of ex vivo and in vivo BMP-2 gene delivery.
2. Please create a table that summarizes advantages and disadvantages of different techniques (virus versus non-viral vectors) used for gene delivery/therapy.
- As per your recommendation, we have added a table summarizing the advantages and disadvantages of viral vs non-viral vectors for BMP-2 gene delivery.
3. For comments 1 and 2, transfection efficiency should also be discussed.
- As per your comments, we have included a discussion on the efficiency of transfection in Tables 1 and 2.
4. Fig. 2a: Abbreviations (B, BM, C, L, PM, WB) should be defined in the figure legends.
- Thank you for your comments. We have added the definitions of the abbreviations.
5. Line 157: Figure 3 related to rhBMP-2 was not included in the manuscript.
- Thank you for your comments. There was an error in the writing, and we have revised the manuscript.
6. Lines 217-218 about lentivirus: References should to be cited. In addition, lentivirus or retrovirus has recently been widely used in the fields, and thus the authors may want to expand the discussion (lines 375-382) further.
- The references have been added to the sentences. The discussion related to lentivirus or retrovirus has been added in the section of viral vectors in in vivo gene delivery.
7. Lines 226-231: Examples related to the use of adeno-associated viral vectors (AAVs) for in vivo gene therapy should be discussed and cited.
- Discussions related to AAV have been added to the section.
8. Lines 305-307 about the use of each of different cell types: References should to be cited.
- Thank you for your comments. The references have been added to the sentences.
9. Fig. 4: Abbreviations (nBMSCs vs dBMSCs) should be defined in the figure legends.
- Thank you for your comments. We have added an explanation about the abbreviations.
10. Minor language editing and proofreading is needed. For example, line 44 – it should be “less functional” instead of “malfunction”.
- Thank you for your comments. We re-purchased the English proofreading service for the revised manuscript.
11. Please review the reference list thoroughly; for example, reference #63 (line 710).
- Thank you for your comments. We have thoroughly checked and revised the reference list.
Round 2
Reviewer 2 Report
Two minor changes:
1. Fig. 1a: WB and BM are not defined.
2. Fig. 5 should be now Fig. 4. Also see Line 333.
Author Response
1. Fig. 1a: WB and BM are not defined.
A) Thank you for your comment. We added explation about the abbreviation at legend of figure 1.
2. Fig. 5 should be now Fig. 4. Also see Line 333.
A) Thank you for your comment. We revised the number of the figure in the manuscript.